# SARS-CoV-2 and the Immune Response in Pregnancy with Delta Variant Considerations

**Patrida Rangchaikul and Vishwanath Venketaraman \***

College of Osteopathic Medicine of the Pacific, Western University of Health Sciences, Pomona, CA 91766-1854, USA; patrida.rangchaikul@westernu.edu
* Correspondence: vvenketaraman@westernu.edu

**Abstract:** As of September 2021, there has been a total of 123,633 confirmed cases of pregnant women with SARS-CoV-2 infection in the US according to the CDC, with maternal death being 2.85 times more likely, pre-eclampsia 1.33 times more likely, preterm birth 1.47 times more likely, still birth 2.84 times more likely, and NICU admission 4.89 times more likely when compared to pregnant women without COVID-19 infection. In our literature review, we have identified eight key changes in the immunological functioning of the pregnant body that may predispose the pregnant patient to both a greater susceptibility to SARS-CoV-2, as well as a more severe disease course. Factors that may impede immune clearance of SARS-CoV-2 include decreased levels of natural killer (NK) cells, Th1 CD4+ T cells, plasmacytoid dendritic cells (pDC), a decreased phagocytic index of neutrophil granulocytes and monocytes, as well as the immunomodulatory properties of progesterone, which is elevated in pregnancy. Factors that may exacerbate SARS-CoV-2 morbidity through hyperinflammatory states include increases in the complement system, which are linked to greater lung injury, as well as increases in TLR-1 and TLR-7, which are known to bind to the virus, leading to increased proinflammatory cytokines such as IL-6 and TNF-$\alpha$, which are already elevated in normal pregnant physiology. Other considerations include an increase in angiotensin converting enzyme 2 (ACE2) in the maternal circulation, leading to increased viral binding on the host cell, as well as increased IL-6 and decreased regulatory T cells in pre-eclampsia. We also focus on how the Delta variant has had a concerning impact on SARS-CoV-2 cases in pregnancy, with an increased case volume and proportion of ICU admissions among the infected expecting mothers. We propose that the effects of the Delta variant are due to a combination of (1) the Delta variant itself being more transmissible, contagious, and efficient at infecting host cells, (2) initial evidence pointing to the Delta variant causing a significantly greater viral load that accumulates more rapidly in the respiratory system, (3) the pregnancy state being more susceptible to SARS-CoV-2 infection, as discussed in-depth, and (4) the lower rates of vaccination in pregnant women compared to the general population. In the face of continually evolving strains and the relatively low awareness of COVID-19 vaccination for pregnant women, it is imperative that we continue to push for global vaccine equity.

**Keywords:** SARS-CoV-2; COVID-19; delta variant; pregnancy; immune response



## 1. Introduction

Ever since the lockdown of Wuhan on 23 February 2020 and the announcement of the global pandemic in March of 2020, the novel coronavirus has claimed over 5 million lives with over 245 million cases [1], and has greatly impacted every individual around the world in all facets of life, not just physically, but also in terms of economic and social effects that may reverberate through generations to come. This enveloped single-stranded RNA virus from the Coronaviridae family, named the severe acute respiratory syndrome coronavirus-2 (SARS-CoV-2) [2], has been greatly researched and reported over the past year, with more and more focus on its impact on each subset of the population. One particular cohort that

is in need of further discussion, and which is unfortunately often excluded from trials, is the pregnant population. Now more than ever, due to the more deleterious effects of the Delta variant on this cohort, as well as the lower rates of vaccination [2], we need greater in-depth discussions on how exactly various changes in the pregnant body may impact the fight against SARS-CoV-2 infection, what the implications are for managing COVID-19 pregnancies, as well as the importance of promoting vaccine awareness to pregnant patients moving forward.

To evaluate how changes in the pregnant body may affect the COVID-19 disease course, it is crucial to first understand how the general population and the normal immune system responds to the virus. SARS-CoV-2 is a single-stranded RNA (ssRNA) virus known to transmit from person to person via respiratory droplets, entering the body via the nasal passage and moving on to infect pulmonary cells [3]. The mechanism of viral entry requires the presence of SARS-CoV glycoprotein receptor angiotensin-converting enzyme 2 (ACE2) [4], which binds to the viral spike protein (S) and is the main determinant of SARS-CoV-2 infectivity. This process is aided through priming via other proteases, including the transmembrane protein serine protease 2 (TMPRSS2), which cleaves the spike protein for fusion into the host cell [5]. It has also been identified that cells co-expressing both ACE2 and TMPRSS2 are the most susceptible to SARS-CoV-2 entry [6].

Once the virus enters the host cell, it is then able to replicate and is subsequently released from the host cell, causing pyrotosis (lytic programmed cell death) in the process. This triggers the release of damage-associated molecular patterns (DAMPs) such as ATP and nucleic acids that go on to cause neighboring cells to initiate an inflammatory response. This response includes the release of IL-6, CXCL10 and type 1 interferons, which are chemoattractants that recruit monocytes, macrophages, and T cells to the site of infection, triggering further proinflammatory signals in a positive feedback loop. For most healthy patients with no comorbidities, the recruitment of CD4+ T-helper 1 cells is adequate to clear the infection, preventing further viral spread and inflammation, eventually leading to full recovery [6].

It is understood that severe COVID-19 cases arise not as a direct effect of viral replication itself, but rather as a result of the host's exaggerated inflammatory response. This excessive inflammation leads to the damage of lung integrity and function [7]. It may also lead to the initiation of a cytokine storm, which may result in acute respiratory distress syndrome (ARDS), multisystem organ failure, and a subsequent increase in mortality risk. In this immune response balance, IL-1, IL-6, IL-18, IFN-$\gamma$, TNF-$\alpha$, and hyperferritinemia are proinflammatory factors pushing the system towards a cytokine storm, with IL-6 elevation being the most reported to correlate with higher mortality [8,9]. On the other hand, protective anti-inflammatory factors such as neutralizing antibodies, type 1 interferons, IL-10, and regulatory T cell expansion are key in reaching the resolution stage [10].

## 2. Implications of Immune Modulation in Pregnancy and SARS-CoV-2

Delicate changes in the maternal immune system increases the risk for severe COVID-19 infection in pregnancy when compared to the general population [11]. This is partly due to the immunological accommodations that need to be made in order to host the semi-allogenic fetus. From previous research thus far, mainly drawing from the works of Wastnedge et al. and Abu-Raya et al., we can identify eight main changes to the maternal immune response and physiology, which may be split into two categories.

### 2.1. Overall Immune Attenuation in Pregnant Physiology

Firstly, there is an overall attenuation of the immune system in the pregnant physiology, increasing pregnant women's susceptibility to SARS-CoV-2 and impairing viral clearance, with specific changes as follows (Figure 1):

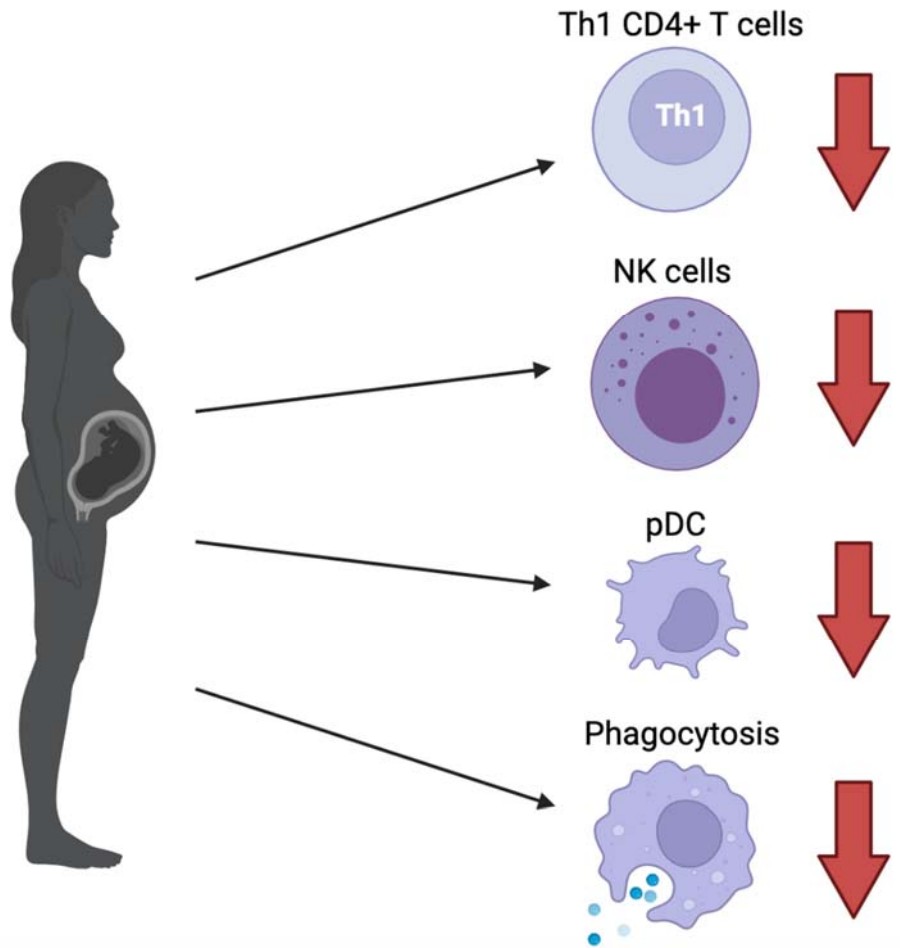

**Figure 1.** General attenuation of various components of the immune system in pregnancy, leading to increased susceptibility to SARS-CoV-2 and decreased efficacy of viral clearance. These include a shift from Th1 to Th2 CD4+ T cells, decreased natural killer (NK) cells, plasmacytoid dendritic cells (pDC) producing type 1 interferons, and the phagocytic index of neutrophil granulocytes and monocytes. In addition to these specific changes, increased progesterone in the maternal circulation further dampens immune response.

(1)  The CD4+ T cell population has been shown to shift from T helper type 1 (Th1) to Th2 predominant during pregnancy. Th1 CD4+ T cells help boost cellular immune response through activating macrophages, cytotoxic T lymphocytes and NK cells through IL-2 and IF-γ signaling, whereas Th2 CD4+ T cells coordinate the humoral response through activating eosinophils, basophils, and mast cells via IL-4 and IL-6 signaling. Usually, in non-pregnant individuals, a robust Th1 response is associated with good prognosis for viral infections [6,12–14]. With the shift from Th1 to Th2 CD4+ T cells in pregnant women, this carries unfavorable implications. For further evidence, Pavel et al. discussed how the Th2/Th1 cytokine imbalance was found to be significantly higher in patients with known COVID-19 risk factors such as age (>40), sex (male), active smoking, as well as ACE2 expression and patients with asthma, leading to higher mortality risk [15].

(2)  NK cells that play a critical role in the innate immune system are shown to decrease in circulation during pregnancy, further impeding the internal battle against COVID-19 exposure [6]. Hsieh et al. demonstrated that the cytolytic effects of NK cell function play an important role in SARS-CoV-2 clearance. In particular, NK cells that expressed receptor DNAM1 are linked to more rapid recovery [16].

(3)  Lampe et al. found a significant decrease in the phagocytic index of neutrophil granulocytes and monocytes in both healthy and pre-eclamptic pregnancies, which

may have implications for hindering maternal clearance of viral infection [17,18]. However, there is currently too limited data to be able to make solid conclusions.

(4) Progesterone is a steroid with immunomodulatory properties that are increased in maternal circulation [19]. In a mouse model of influenza A infection, this has shown to decrease virus-specific antibody levels as well as virus-specific CD8+ T cells. Upon re-challenging with influenza A, this resulted in a more severe disease course [20].

(5) Plasmacytoid dendritic cells (pDCs) are key for type 1 interferon production against viruses and are also decreased in the maternal circulation [6,21,22]. Additionally, pDCs from pregnant women were reported to have attenuated the inflammatory response to the H1N1/09 virus, which is thought to be one of the reasons why pregnant women were more severely affected [23]. However, it has been recently shown that a robust type 1 interferon response is associated with hyperinflammation and severe COVID-19 infection, as opposed to a more delayed and possibly suppressed interferon response in early infection. This speaks to the importance of understanding the different roles of type 1 interferon at each stage of infection for therapeutic decisionmaking [24]. In previous studies on SARS-CoV and MERS-CoV, type 1 interferons are known to decrease the expression of IFN receptors, leading to a systemic inflammatory response [25]. In the context of pregnancy, what has been discussed thus far suggests that decreased pDCs in maternal circulation are unfavorable for earlier stages of infection when viral clearance is key, however it may play a protective role against the development of a cytokine storm in the later stages of infection.

The immune response level also changes as the pregnancy progresses. Abu-Raya et al. have demonstrated how the level of T cells and NK cells decrease mostly in the first trimester and plateaus throughout the second and third trimester. This may point to unfavorable outcomes earlier in pregnancy with SARS-CoV-2 infection. On the other hand, the Th1 CD4+ T cell response, as well as the rate of phagocytosis, begins to decrease slightly in the first trimester and continues to decrease rapidly in the second and third trimesters. This suggests a greater susceptibility to SARS-CoV-2 and a worse prognosis for expecting mothers infected in the third trimester [26].

### 2.2. Immune Changes in Pregnancy Leading to Hyperinflammation in SARS-CoV-2

This leads us to the second category of immune changes in the pregnant body that may exacerbate hyperinflammatory states, increasing the risk for acute lung injury and diseasing morbidity in the face of SARS-CoV-2 infection (Figure 2):

(1) Previous studies have shown an increase in maternal serum levels of C3a, C4a, C5a, C4d, C3, C9, and the Serum Complement Membrane Attack Complex SC5b9 when compared to non-pregnant women [27,28]. This increase in complement activation is linked to greater lung injury and disease severity in SARS-CoV-2 infection [29]. Elevation of C3 activation products was observed in the lung as early as 1 day postinfection. In C3 deficient mice (C3−/−), SARS-CoV-2 infection demonstrated reduced weight loss and respiratory dysfunction with an equivalent viral load, as well as significantly less neutrophils and inflammatory monocytes [30]. Gralinski et al. further proposed the attenuation of the complement system as a possible effective treatment option for SARS-CoV-2 [30].

(2) Young et al. from the *American Journal of Obstetrics and Gynecology* previously showed an increase in IL-6, IL-12, IFN-$\alpha$, and TNF-$\alpha$ in the maternal sera in uncomplicated pregnancies when compared to nonpregnant controls, with IL-12 remaining elevated into the postpartum period [31]. As previously discussed, an increase in IL-6 is especially correlated with higher mortality in SARS-CoV-2 infection, suggesting an already vulnerable physiological state for expecting mothers.

(3) The role of toll-like receptors is also crucial in this discussion. Young et al. additionally reported elevated levels of TLR-1, TLR-7, and TLR-9 when compared to nonpregnant values in women [31]. This has several implications. Firstly, SARS-CoV-2 spike protein has been shown to bind to TLR-1, as well as TLR-4 and TLR-6, suggesting

another mechanism that may increase pregnant women's susceptibility to COVID-19 infection [32]. TLR-7 is expressed on monocyte-macrophages and dendritic cells, and are important in the recognition of ssRNA viruses such as SARS-CoV-2 [33]. Whole genome sequencing showed that TLR-7 has more ssRNA motifs that can bind to SARS-CoV-2 when compared to SARS-CoV and MERS-CoV [1]. Binding of the S glycoprotein on the surface of the viral envelope to ACE2 may be recognized by TLR-7, leading to an increased production of IL-1, IL-6, monocyte chemoattractant protein-1 (MCP-1), MIP-1A, TNF-α, and type 1 interferons [34], which could lead to a hyperinflammatory state and acute lung injury [25].

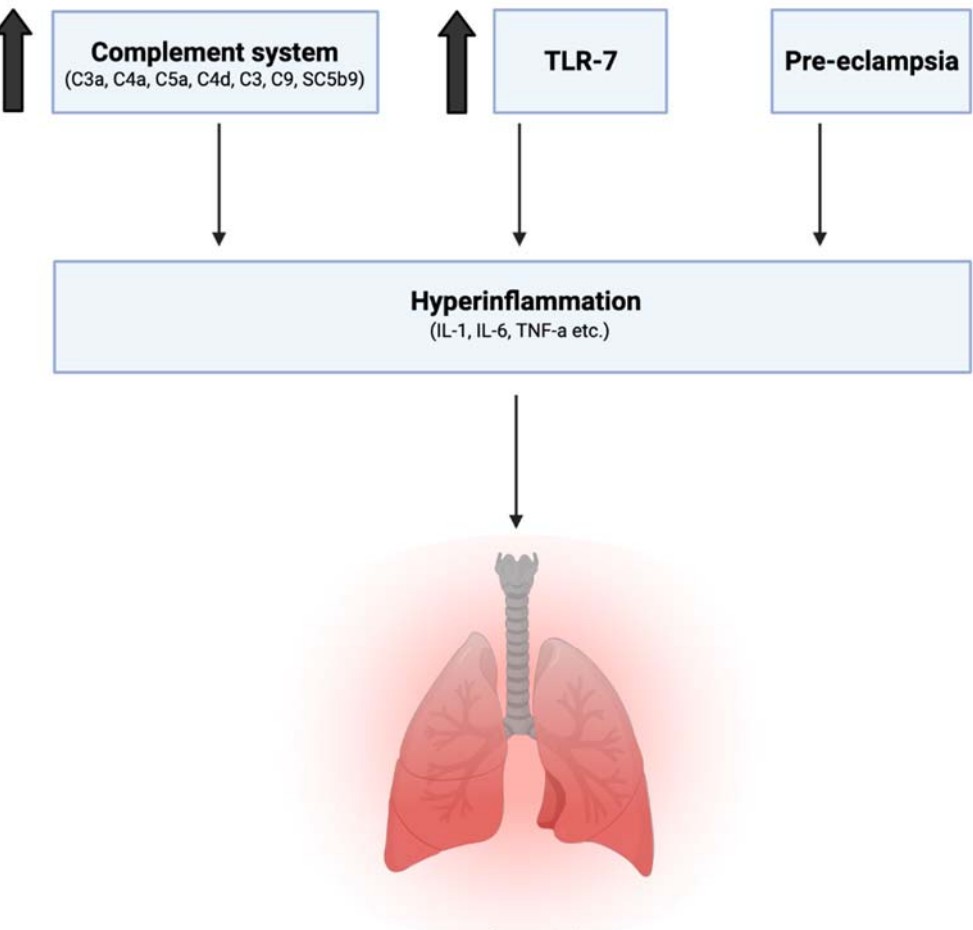

**Figure 2.** Immune changes in pregnant physiology that increase risk of hyperinflammation in SARS-CoV-2 infection: Increase in complement activation, toll-like receptors TLR-1 and TLR-7, as well as the preeclamptic state, has been linked to the release of pro-inflammatory factors and greater lung injury in SARS-CoV-2. This is on top of increased levels of IL-6, IL-12, IFN-α, and TNF-α already observed in normal pregnancy when compared to nonpregnant levels.

## 3. Implications of Other Pregnancy-Specific Physiological Changes and SARS-CoV-2

### 3.1. Angiotensin Converting Enzyme

Another factor that can predispose the pregnant body to greater SARS-CoV-2 susceptibility is the elevation of angiotensin converting enzyme 2 (ACE2) in the maternal circulation [17,35]. Since SARS-CoV-2 spike proteins bind onto angiotensin converting enzymes (ACE) for entry into the host cell, this suggests a higher probability of ACE2 expression on the surface of pneumocytes, leading to increased viral binding, entry, and continuation of pathogenesis [36] (Figure 3).

**Figure 3.** ACE2 and TLR-1 are known to bind to SARS-CoV-2 spike protein S1 and are increased in maternal circulation, leading to increased susceptibility of pregnant women to SARS-CoV-2 infection through increased viral binding and subsequent fusion into host cell.

*3.2. Human Leukocyte Antigen (HLA)*

An important topic to mention is the role of HLA genes in pregnancy when hosting the semi-allogenic fetus and its effects on the maternal immune response. Immunologically tolerating the fetus is an essential phenomenon in pregnancy, of which HLA-C and HLA-G are some of the important genes involved in establishing this maternal-fetal immune intolerance [14,37]. Firstly, paternal HLA-C has been shown to regulate CD4+ and CD8+ T lymphocyte activation through the expression of IL-10 and TGF-β [14]. Secondly, HLA-G is another crucial gene detected only in fetal cells at the maternal-fetal junction, unlike other class I genes, which are expressed on nearly all nucleated cells [37]. The HLA-G isoforms contain a transmembrane a1-domain that is known to be capable of inhibiting NK cell activity, which is thought to play a role in preventing trophoblast cell destruction by NK cells in the uterus, where they are abundant [37]. HLA-G has also been shown to induce T lymphocyte death [38]. When considered together, the immunomodulatory effects of HLA-C and HLA-G in pregnancy may have further implications in the face of SARS-CoV-2 infection and possible increased susceptibility. However, since studies so far have mainly focused on the maternal–fetal interface, it is still unclear what the effects of HLA genes in pregnancy are on the systemic immune response. This may be a future area of exploration in investigating pregnancy state susceptibility to SARS-CoV-2 infection.

### 3.3. Pre-Eclampsia in SARS-CoV-2

Pre-eclampsia is a phenomenon in pregnancy defined by systemic endothelial cell dysfunction. It is theorized to be caused by abnormal placentation where there is abnormal blood flow to the placenta, leading to placental hypoxia [39]. This triggers inflammation and the release of antiangiogenic and inflammatory factors into the circulation, thereby causing the systemic endothelial cell dysfunction. This process leads to decreased endothelial cell production of prostaglandins and vasodilators, causing systemic vasoconstriction and thus hypertension [40]. In relation to SARS-CoV-2 infection, the endothelial cell dysfunction in pre-eclampsia prevents the blockage of immune cell entry into pulmonary cells, leading to excessive inflammation in the lungs, possibly progressing to ARDS [6].

Several studies have shown that pre-eclampsia is associated with increased levels of inflammatory signals, as well as lower levels of immune regulatory markers. A study in Mexico showed significantly decreased levels of regulatory T cells (Tregs) in the serum of women with differing severities of pre-eclampsia. Increased levels of inflammatory signals such as IL-6, TNF-$\alpha$, IL-8, CXCL10, CCL2, and CXCL9 were also observed in those severely pre-eclamptic [41]. Another meta-analysis concluded that there are significantly decreased IL-10 levels observed in those with pre-eclampsia at the time of diagnosis [42]. As previously discussed, resolution from SARS-CoV-2 infection requires the reduction of inflammatory signals such as IL-6 and TNF-$\alpha$ by protective mediators such as regulatory T cells (Treg) and IL-10. Therefore, it is consistent with this data that we observe higher rates of hospitalization from COVID-19 amongst pre-eclamptic pregnant women [43].

### 3.4. Coagulation in SARS-CoV-2

Another physiologic change the body adapts to during pregnancy is an increase in fibrinogen and D-dimer levels, inducing a hypercoagulable state, especially during the third trimester [44]. The purpose of this physiological change is thought to be to reduce maternal bleeding during delivery. Normally, D-dimer levels would be considered elevated if above 500 ng/mL [45]. However, during pregnancy, reference ranges can reach over 4000 ng/mL in the third trimester after adjusting for risk factors [44]. Several studies have shown a correlation between D-dimer levels as a prognostic hematological marker for COVID-19 disease severity [46,47]. The implications of this in relation to the already elevated D-dimer values in pregnancy is still unclear.

When considered in the context of COVID-19, severe SARS-CoV-2 infection may lead to coagulopathy due to factors such as increases in angiotensin II, decreases in vasodilator angiotensin, and the release of cytokines and other inflammatory responses [46]. Up to 50% of patients with severe SARS-CoV-2 infection report coagulopathy [46]. The evidence points to the possibility of even more severe hematological complications in pregnant patients with COVID-19, who already have a hypercoagulable state to begin with, as well as the importance of anticoagulation, especially when treating these pregnant women. Current guidelines suggest giving systemic anticoagulation to all hospitalized pregnant patients diagnosed with SARS-CoV-2 until 10 days postnatal [48,49]. Unfractionated heparin, low molecular weight heparin, and warfarin are deemed safe in pregnancy and also do not accumulate in breast milk. However, use of direct-acting oral anticoagulants such as apixaban are not routinely recommended in pregnancy due to a lack of safety data [49]. Furthermore, questions have arisen regarding the use of transexamic acid (TXA) or hemabate as prothrombotic factors for managing postpartum hemorrhage, especially since hemabate is contraindicated in patients with asthma due to bronchospasm. However, since hemabate is usually given in the context of viral pneumonia, the American College of Obstetrics and Gynecology recommends continued use of both thrombotic medications as normally indicated, while keeping in mind the already increased risk of clotting due to COVID-19 infection state [50].

It is also still unclear how hyperferritinemia in severe COVID-19 infections can affect the pregnant patient. Since hyperferritinemia may exacerbate coagulation [51], this may also lead to further hematological complications. However, since anemia is usually a

manifestation of pregnancy due to an increased volume state, our understanding is that hyperferritinemia by itself may be less likely to predispose the pregnant patient to an increased risk of thrombosis.

## 4. COVID-19 Epidemiological Data and Clinical Concerns in Pregnancy Thus Far

### 4.1. Overview

After discussing the multitude of nuanced immunological and physiological changes that occur in the pregnant body in the face of SARS-CoV-2 infection, we can now analyze current epidemiological and clinical data with a greater understanding. According to the CDC's most recent September 2021 analysis on the impact of COVID-19 on pregnancy since the start of the pandemic, there has been a total of 123,633 cases of pregnant patients with confirmed SARS-CoV-2 infection in the US [11]. Compared to non-pregnant women of reproductive age with COVID-19 infection, ICU admission was 2.13 times more likely in the pregnant population, invasive ventilation 2.59 more likely, and extracorporeal membrane oxygenation (ECMO) 2.02 times more likely [52]. Compared to pregnant women without COVID-19 infection, maternal death was 2.85 times more likely in those infected, pre-eclampsia 1.33 times more likely [53], preterm birth 1.47 times more likely, stillbirth 2.84 times more likely, and NICU admission 4.89 times more likely [11]. The data presented thus far in the obstetric literature corroborates strong evidence pointing to SARS-CoV-2 infection having a more severe impact on both pregnant women and neonatal outcomes.

### 4.2. Vertical Transmission

A concern that has been in discussion ever since the start of the pandemic is the likelihood of SARS-CoV-2 vertical transmission [54]. The literature consensus seems to be that vertical transmission from the mother to the neonate is possible but only occurs in a minority of cases in the third trimester of pregnancy [55]. A systemic review and meta-analysis conducted by Kotlyar et al. from the *American Journal of Obstetrics and Gynecology* (AJOG) in January 2021 included 30 case reports and 38 cohort or case series studies testing 979 neonates with a viral RNA test in mothers who received a COVID-19 diagnosis, which showed a 3.2% rate of vertical transmission among pregnant women infected in the third trimester [55]. The CDC also reported similar results, showing that, amongst 25,896 neonates born to pregnant women with COVID-19, 3381 (13%) underwent PCR testing, of which 136 (4%) were PCR positive. Of those that tested positive, almost all were born to mothers who contracted infection within 2 weeks of birth, with greater positivity in those born preterm [11]. The low rate of vertical transmission seen is consistent with transcriptomic data showing that it is rare for placental cells to co-express both the ACE2 and TMPRSS2 proteins that are required for SARS-CoV-2 viral entry into the cell [56].

There have also been questions on whether or not vaginal deliveries increase the risk of vertical transmission, as opposed to a caesarian section. A systematic review by Cai et al. published in February 2021 showed no significant difference in neonatal infection rate and that there is insufficient data to support caesarian deliveries as a safer alternative at this time [57].

### 4.3. Treatment and Delivery Protocol

For a more complete clinical picture, we also touch on special considerations for the treatment of SARS-CoV-2-positive pregnant patients. Since the start of the pandemic, Remdesivir was the first drug approved for the treatment of COVID-19. This nucleoside analog drug prevents viral replication and has been shown to reduce the length of hospitalization in patients requiring oxygen therapy [58]. However, there is still limited data on the use of Remdesivir in pregnant COVID-19 patients, and there are no clear guidelines as of yet. All clinical trials for the use of Remdesivir in COVID-19 patients excluded pregnant and breastfeeding patients [59]. However, one study reported the use of Remdesivir requested by physicians through the compassionate use program on 86 hospitalized pregnant patients with confirmed SARS-CoV-2 and oxygen saturation ≤94%. On day 28 post admin-

istration of Remdesivir, oxygen requirement decreased in 96% of pregnant patients and in 89% of postpartum patients. Of the pregnant patients, 93% on mechanical ventilation were extubated, 93% recovered, and 90% were discharged. The study reported that Remdesivir was well tolerated overall, with severe adverse events in 16% of the cases, one maternal death, and no neonatal deaths [60]. However, there is still a lack of equitable inclusion of pregnant and breastfeeding patients for COVID-19 treatment trials and a lack of data for making clear clinical guidelines.

Along with Remdesivir, systemic corticosteroids have also been indicated in critically ill patients with mixed results. A meta-analysis of seven randomized controlled trials investigating corticosteroid use in 851 nonpregnant patients with ARDS showed a reduction in all-cause mortality when compared to placebo [61]. The RECOVERY trial, which included 6425 patients, similarly demonstrated a reduction in the mortality of hospitalized patients requiring supplemental oxygen when given dexamethasone for 10 days. No benefit was seen in patients who did not require supplemental oxygen [62].

In the context of pregnancy, since antenatal steroids are already routinely used for fetal lung maturity in preterm births (which are already more common in COVID-19 positive pregnancies) [63], and with the potential benefit of decreased maternal mortality, the National Institutes of Health (NIH)'s *COVID-19 Treatment Guidelines* recommends a short course of dexamethasone for hospitalized pregnant patients who require supplemental oxygen, with or without mechanical ventilation [64]. Since current recommendations suggest a continuation of dexamethasone for up to 10 days or until hospital discharge, further discussion should be made in the context of gestational diabetes and controlling hyperglycemic episodes. A small randomized controlled trial demonstrated fewer hyperglycemic episodes in patients who received 12 mg every 12 h for 1 day when compared to those who received 6mg every 12 h for 2 days, suggesting that higher doses for a shorter period of time is more favorable for the metabolic profile of pregnant patients with gestational diabetes receiving systemic corticosteroids. However, data are still limited [65].

The American College of Obstetrics and Gynecology (ACOG) recommends postponing delivery for pregnant patients with suspected or confirmed SARS-CoV-2 infection that have recovered until a negative test result is obtained. Otherwise, the timing of delivery should not be dictated by infection status. The decision of caesarean section versus vaginal delivery, as well as delayed cord clamping, should also not be affected by the mother's infection status [50]. For SARS-CoV-2-positive mothers, the ACOG additionally recommends strict use of PPE adherent to CDC guidelines at the time of delivery.

Postpartum considerations have led to the discussion of whether or not the neonate should be kept in the same room as a COVID-19-positive mother. The ACOG notes that the benefits of early close contact between mother and neonate, including increased successful latching and the facilitation of family bonding, should be considered in comparison to the risks of late-onset neonatal infection [66]. Although some data demonstrates an increased risk, other studies show there is no difference in the risk of neonatal COVID-19 infection when the neonate is placed in the same room as the mother or when cared for separately [67].

*4.4. Breastfeeding*

According to the CDC, breast milk is not likely to be a source of infection. Although breast milk may test positive for SARS-CoV-2 due to residual genetic material, there is no capability for viral replication to occur. However, since droplets may still be transferred between the COVID-19 positive mother and the neonate, the ACOG recommends breast pumping while observing proper hand hygiene [66]. Furthermore, in terms of breastfeeding benefits, SARS-CoV-2-specific IgA and IgG have been detected in the breastmilk of 22 COVID-19-vaccinated (Pfizer/Moderna) individuals [68,69].

## 5. Delta (B.1617.2) and Other Variant Considerations

### 5.1. Concerning Data on the Impact of the Delta Variant on Pregnancies

Towards the end of June 2021, the CDC reported a 7-day moving average of approximately 12,000 reported cases. On 27 July, the number of cases increased to over 60,000 throughout the country with an alarming rise in hospitalization rates [70]. At this time, new data emerged describing a new strain of the coronavirus named the "Delta" variant which has now been labeled as a "Variant of Concern (VoC)". The CDC has described this particular strain to be over twice as contagious as the previous variants, leading to greatly increased transmissibility between individuals. A retrospective cohort study in Canada found that patients infected with the Delta variant had a 108% increased risk of hospitalization compared to previous strains, 234% for ICU admission, and 132% for death [71].

The Delta variant surge has more severely impacted the pregnant population as well. On September 13, the *American Journal of Obstetrics and Gynecology* published a manuscript reporting "Increasing severity of COVID-19 in pregnancy with Delta variant surge" from the University of Texas Southwestern Medical Center. The article reports that "as the Delta (B.1.617.2) variant predominated locally, both the case volume and the proportion of severe or critical illness increased significantly ($p = 0.001$)" and that over a quarter of the pregnant patients diagnosed between August 29 and September 4 of 2021 required hospital admission for severe illness. The greatest morbidity was seen in underserved populations where rates of vaccinations are lowest. The article also pointed out that those potential pathophysiologic mechanisms for an increased severity in illnesses with B.1.617.2 during pregnancy are unclear [72]. Similar reports have also surfaced: The University of Alabama at Birmingham hospital observed an increase in ICU admission from 8% to 29% between the non-Delta and Delta cohorts, as well as an increase in the rates of cesarean delivery preterm birth and NICU admission. All patients admitted to the hospital were unvaccinated [2].

### 5.2. Possible Pathophysiologic Explanations for the Increased Morbidity of Delta Variant on Pregnant Patients

In our discussion of possible explanations of why the Delta variant is causing greater morbidity and hospitalization rates amongst pregnant women, we believe that it is a combination of: (1) The Delta variant itself being more transmissible, contagious, and efficient at infecting host cells; (2) The initial evidence pointing to the Delta variant causing a significantly greater viral load that accumulates more rapidly in the respiratory system [73]; (3) The pregnancy state being more susceptible to SARS-CoV-2 infection, as previously discussed in depth; (4) The lower rates of vaccination in pregnant women compared to the general population.

Firstly, the current understanding of what makes the Delta variant significantly more contagious focuses on one particular mutation on the SARS-CoV-2 spike protein called "P681R", where a proline residue is substituted for arginine in the furin cleavage site where host enzymes can effectively cut the protein, essentially activating the viral particles that can immediately go on to infect other cells [74]. Although the alpha variant also contains a mutation at this site, Liu et al. demonstrated a significantly reduced spike protein cleavage time in the Delta variant when compared to the alpha variant, and that, when the P681R mutation was eliminated, Delta's infective advantage was nullified [54]. On top of an efficiency in cleavage, it has also been shown that the Delta variant excels at cell fusion—an essential step in infectivity, with almost triple the rate [75]. However, since the Kappa mutation also carries the same P681R mutation, but has not had the same deleterious effects, it is understood that there are more mutations at play in the Delta variant of concern [76].

On this topic of other possible mutations at play, when specifically focusing on increased morbidity and proportions of ICU admissions with the Delta variant observed in pregnancy, one theory could be that the Delta variant mutations may contain ssRNA motifs with greater binding affinity to toll-like receptors such as TLR-1 and TLR-7. As

previously discussed, these specific TLRs have been shown to increase in pregnancy, with the spike–protein binding to TLR-1 for host entry, and TLR-7 recognizing spike–protein binding to ACE2 on the host cell leading to increased IL-6 and an increased production of other proinflammatory cytokines, subsequently causing hyperinflammation, acute lung injury, and an overall increase in the severity of the disease. There is more yet to be uncovered with the Delta variant and, with the prospect of other variants that may emerge, these discussions are ongoing.

### 5.3. Anticipation of Future Variants

Delta variant aside, the CDC continues to monitor other variants, and the virus continues to evolve. The SARS-CoV-2 virus genome has been shown to have around 30,000 nucleotides, with the first complete genome sequencing of the prototype HCoV-OC43 strain having 30,738 nucleotides [77], with an evolutionary rate estimation of $1.12 \times 10^{-3}$ to $6.25 \times 10^{-3}$ nucleotide substitutions per site per year [78]. Although most RNA viruses have a higher mutation rate than DNA viruses, the coronaviruses have a proofreading mechanism that explains why the SARS-CoV-2 virus has a slower mutation rate than the influenza viruses [79].

Most recently, it has been reported that a newly-discovered mutation of the Delta variant, dubbed the "Delta plus" (AY.4.2), has emerged, reported to have a greater proportion of mutations on its spike protein (over 20% more common than in the original Delta variant) [80], and is currently under investigation in the U.K. with the concern that this mutation could make the already more contagious Delta variant even more transmissible, potentially undermining the current COVID-19 vaccines. On 20 October 2021, it was classified as a "variant under investigation" by the WHO and is still being closely monitored [81].

### 6. Vaccine Considerations for Pregnancy

In the face of new variants that may emerge, it is not unreasonable to suggest that the human immune system naturally selects more and more contagious strains of the virus, and that, over time, the virus will become increasingly efficient at infecting the host. It is therefore imperative to promote for vaccine equality throughout the world so as to prevent the emergence of new strains with greater transmissibility and virulence that may continue to undermine our current vaccines, spreading through borders from one country to another. We can only hope that our current situation will not progress to a point where even vaccinated individuals are no longer well protected.

This, of course, holds true for pregnant individuals as well. Many expecting mothers are still hesitant to receive COVID-19 vaccinations despite mounting evidence on its benefits for not only the mother, but for the neonate as well. A cohort study published in the *American Journal of Obstetrics and Gynecology* in March of 2021 investigating the COVID-19 vaccine response in 131 pregnant and lactating women found that the vaccine-generated antibodies were present in all umbilical cord blood and breastmilk samples, with significant evidence supporting higher antibody titers generated from the vaccine when compared to those generated from active infection ($p < 0.0001$). Results also showed that vaccine-induced antibody titers were equivalent in both pregnant women and non-pregnant women [57].

### 7. Summary

In summary, the eight key changes that we have identified can be put into two categories: Factors that may impede immune clearance of SARS-CoV-2 include decreased levels of NK cells, Th1 CD4+ T cells, and plasmacytoid dendritic cells, a decreased phagocytic index of neutrophil granulocytes and monocytes, as well as the immunomodulatory properties of progesterone, which is elevated in pregnancy. Secondly, factors that may exacerbate SARS-CoV-2 morbidity through hyperinflammatory states include increases in the complement system, which are linked to greater lung injury, as well as increases in TLR-1 and TLR-7, known to bind to the virus, and leading to increased pro-inflammatory

cytokines such as IL-6 and TNFα, which are already elevated in normal pregnant physiology. An increase in ACE2 may also lead to increased viral binding on spike proteins, as previously discussed.

We also proposed that the increased case volume and proportion of ICU admissions of pregnant women due to the Delta variant may be explained from a combination of (1) the Delta variant itself being more transmissible, (2) initial evidence pointing to the Delta variant causing a significantly greater viral load that accumulates more rapidly in the respiratory system, (3) the pregnancy state being more susceptible to SARS-CoV-2 infection, and (4) the lower rates of vaccination in pregnant women compared to the general population. At a molecular level, it could also be that the Delta variant contains other mutations that we do not yet know, which produces consequences we are not yet aware of. For example, there could be an ssRNA motif with greater binding affinity to toll-like receptors such as TLR-1 and TLR-7 which, as previously discussed, have been shown to increase in pregnancy and play some role in spike–protein binding for host entry. This may be an avenue for future research and further exploration.

With this in-depth discussion of the various evidence pointing to multiple mechanistic changes undermining the immune system in pregnancy in ways that increases susceptibility to SARS-CoV-2 infection, impairs efficacy of viral clearance, as well as predisposes the pregnant body to hyperinflammation and worse COVID-19 outcomes, we hope there will be greater efforts to promote vaccination in pregnant women throughout the world, especially in lieu of the Delta variant causing an increase in case volume and the percentage of those hospitalized.

The CDC's conclusion was that "COVID-19 vaccination is recommended for all people aged 12 years or older, including people who are pregnant, breastfeeding, or who are trying to get pregnant now or might become pregnant in the future" and is recommended at all trimesters, at any stage of pregnancy [11]. Any of the FDA-approved COVID-19 vaccines are authorized for use in pregnant women, including Pfizer-BioNTech, Moderna, and Janssen [82]. Unfortunately, the percentage of fully vaccinated pregnant women remains low, with only 30.1% coverage in mothers aged 18–49 years prior to and during pregnancy, according to data gathered between 14 December 2020 and 11 September 2021 [11].

With this data, it is even more urgent that clinicians and health organizations properly communicate the importance of vaccination before, during, and at any stage of pregnancy to all women who wish to become or are currently pregnant. Many are still unaware of this fact and efforts to promote vaccination in pregnant women need to continue to be pushed further for the health of both the mother and the baby.

**Author Contributions:** P.R. and V.V. have contributed to drafting this review. V.V. conceived the framework, provided guidance and assistance, and made edits to the draft. All authors have read and agreed to the published version of the manuscript.

**Funding:** We appreciate the funding support from National Institutes of Health (NIH) award RHL143545-01A1.

**Institutional Review Board Statement:** Not applicable.

**Informed Consent Statement:** Not applicable.

**Data Availability Statement:** Data sharing not applicable. No new data were created or analyzed in this study. Data sharing is not applicable to this article.

**Acknowledgments:** We appreciate the funding support from NIH (RHL143545-01A1).

**Conflicts of Interest:** The authors declare no conflict of interest.

## Abbreviations

| | |
|---|---|
| (SARS-CoV-2) | Severe acute respiratory syndrome coronavirus 2 |
| (MERS-CoV) | Middle East respiratory syndrome |
| (COVID-19) | Coronavirus disease 2019 |
| (ARDS) | Acute respiratory distress syndrome |
| (ICU) | Intensive care unit |
| (NICU) | Neonatal intensive care unit |
| (ECMO) | Extracorporeal membrane oxygenation |
| (ssRNA) | Single-stranded RNA |
| (NK cell) | Natural killer cell |
| (pDC) | Plasmatoid dendritic cell |
| (Th) | T helper |
| (Treg) | Regulatory T cell |
| (TLR) | Toll-like receptor |
| (IFN) | Interferon |
| (IL) | Interleukin |
| (MCP-1) | Monocyte chemoattractant protein-1 |
| (MIP-1$\alpha$) | Macrophage inflammatory protein-1 alpha |
| (TNF-$\alpha$) | Tumor necrosis factor alpha |
| (ACE2) | Angiotensin converting enzyme 2 |
| (ACOG) | American College of Obstetrics and Gynecology |
| (AJOG) | American Journal of Obstetrics and Gynecology |
| (CDC) | Centers for Disease Control and Prevention |
| (FDA) | Food and Drug Administration |
| (NIH) | National Institutes of Health |

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
