# Peer review of "SARS-CoV-2 and the Immune Response in Pregnancy with Delta Variant Considerations"

_2036-7449, doi:10.3390/idr13040091_

Round 1

Reviewer 1 Report

It is a very comprehensive and welcomed review about SARS-CoV-2 and the immune response in pregnancy.

I have some minor suggestions in order to improve the manuscript :

  • Do different immune responses in pregnancy depend on the trimester of pregnancy? Second, Ho to explain the unfavourable evolution of COVID-19 for 5 weeks pregnant woman without comorbidities?
  • I suggest adding reference numbers lines 88, 263, 267
  • Lines 219-224- systematic anticoagulation for all COVID-19 pregnant patients?
  • For COVID-19 vaccination in pregnancy, I suggest mentioning the authorized vaccines? The current WHO guidelines recommend the vaccination of pregnant women against COVID-19, particularly if they are at high risk of exposure or have comorbidities that enhance the risk of severe disease.
  • I suggest a conclusion about the immune response in pregnancy and SARS-Co-V 2 instead of vaccination.
  • I am not sure that all references were cited in the text.

Author Response

REVIEW COMMENTS:

REVIEWER#1

It is a very comprehensive and welcomed review about SARS-CoV-2 and the immune response in pregnancy.

I have some minor suggestions in order to improve the manuscript :

  • Do different immune responses in pregnancy depend on the trimester of pregnancy? Second, How to explain the unfavourable evolution of COVID-19 for 5 weeks pregnant woman without comorbidities?

This is now added to the manuscript:

The immune response level also changes as the pregnancy progresses. Abu-Raya et al. demonstrates how the level of T cells and NK cells decrease mostly in the first trimester and plateaus throughout the second and third trimester. This may point to unfavorable outcomes earlier in pregnancy with SARS-CoV-2 infection. On the other hand, the Th1 CD4+ T cell response, as well as the rate of phagocytosis begins to decrease slightly in the first trimester and continues to decrease rapidly in the second and third trimesters. This suggests a greater susceptibility to SARS-CoV-2 and a worse prognosis for expecting mothers infected in the third trimester [78].

  • I suggest adding reference numbers lines 88, 263, 267

These references have now been added.

  • Lines 219-224- systematic anticoagulation for all COVID-19 pregnant patients?

This paragraph has now been edited as follows:

Current guidelines suggest giving systemic anti-coagulation to all hospitalized pregnant patients diagnosed with SARS-CoV-2 until 10 days postnatal [26, 79]. Unfractionated heparin, low molecular weight heparin, and warfarin are deemed safe in pregnancy and also do not accumulate in breast milk. However, use of direct-acting oral anticoagulants such as apixaban are not routinely recommended in pregnancy due to lack of safety data [79].

  • For COVID-19 vaccination in pregnancy, I suggest mentioning the authorized vaccines? The current WHO guidelines recommend the vaccination of pregnant women against COVID-19, particularly if they are at high risk of exposure or have comorbidities that enhance the risk of severe disease.

FDA authorized vaccines are now included in the vaccination section.

  • I suggest a conclusion about the immune response in pregnancy and SARS-Co-V 2 instead of vaccination.

The conclusion is now edited to incorporate a summary of the immune response in pregnancy and SARS-CoV-2.

  • I am not sure that all references were cited in the text.

We have now gone through the references and made sure every reference was cited. Some of the earlier references were not cited in the text previously.

Reviewer 2 Report

In this review, Rangchaikul, P and Venketaraman, V discuss the immune response to SARS-CoV-2 during pregnancy.  The authors discuss how pregnancy impacts the immune response to the virus.  The authors discuss coagulation and other hematological complications due to SARS-Cov-2 infection.  Furthermore, the authors discuss the potential impact of viral variants arising and how this may impact pregnant women.  Overall, the manuscript is well written and easy to follow, below are minor critics:

  • 1- The authors propose that CD4 T cells are involved in susceptibility to disease induced by viral infection. Modulation of CD4 Th1 response, for example to Th2/Treg phenotype, is a plausible explanation for susceptibility to severe disease or ineffective memory response, but the authors may want to expand on how the shift to a humoral response may not be advantageous in this scenario.  This is based on the premise that an anti-SARS-CoV-2 antibody response would be beneficial for protection- so clarification would be helpful.
  • The authors have multiple figures in the manuscript but do not actually refer to the figures within the text. This would help prompt the reader to look at the figures.
  • The authors discuss how pregnancy impacts T cell responses. A short description of what is known about the cellular response in non-pregnant individuals compared to pregnant women would be helpful since current data show both CD4 and CD8 T cell responses occur in most patients infected by SARS-CoV-2 within 1-2 weeks after symptom onset and produce mainly Th1 cytokines.  The frequency of CD4 T cells targeted to Spike correlates with neutralizing antibody titers (Grifoni et al. Cell 2020). 
  • One immunological component not discussed is the role of classical and non-classical human leukocyte antigen (HLA) genes. These genes are central mediators of an immune response to not only a virus but also in pregnancy.  Some discussion about how these immune response genes may be influencing disease would be of great interest or at least mentioning the potential role of these genes in pregnancy and viral response would be an important element for discussion.

Author Response

REVIEWER#2

In this review, Rangchaikul, P and Venketaraman, V discuss the immune response to SARS-CoV-2 during pregnancy.  The authors discuss how pregnancy impacts the immune response to the virus.  The authors discuss coagulation and other hematological complications due to SARS-Cov-2 infection.  Furthermore, the authors discuss the potential impact of viral variants arising and how this may impact pregnant women.  Overall, the manuscript is well written and easy to follow, below are minor critics:

  • 1- The authors propose that CD4 T cells are involved in susceptibility to disease induced by viral infection. Modulation of CD4 Th1 response, for example to Th2/Treg phenotype, is a plausible explanation for susceptibility to severe disease or ineffective memory response, but the authors may want to expand on how the shift to a humoral response may not be advantageous in this scenario.  This is based on the premise that an anti-SARS-CoV-2 antibody response would be beneficial for protection- so clarification would be helpful.
  • The authors discuss how pregnancy impacts T cell responses. A short description of what is known about the cellular response in non-pregnant individuals compared to pregnant women would be helpful since current data show both CD4 and CD8 T cell responses occur in most patients infected by SARS-CoV-2 within 1-2 weeks after symptom onset and produce mainly Th1 cytokines.  The frequency of CD4 T cells targeted to Spike correlates with neutralizing antibody titers (Grifoni et al. Cell 2020). 

We have grouped these 2 points together since we believe they are of a similar vein and hopefully this newly added paragraph helps to clarify

Th1 CD4+ T cells help boost cellular immune response through activating macrophages, cytotoxic T lymphocytes and NK cells through IL-2 and IFN-g signaling, whereas Th2 CD4+ T cells coordinate the humoral response through activating eosinophils, basophils, and mast cells via IL-4 and IL-6 signaling. Usually, in non-pregnant individuals, a robust Th1 response is associated with good prognosis for viral infections [57, 59, 60, 81]. With the shift from Th1 to Th2 CD4+ T cells in pregnant women, this carries unfavorable implications. For further evidence, Pavel et al. discussed how the Th2/Th1 cytokine imbalance was found to be significantly higher in patients with known COVID-19 risk factors such as age (>40), sex (male), current smoking, as well as ACE2 expression and patients with asthma, leading to higher mortality risk [80].

  • The authors have multiple figures in the manuscript but do not actually refer to the figures within the text. This would help prompt the reader to look at the figures.

Figure referrals are now added to the text.

  • One immunological component not discussed is the role of classical and non-classical human leukocyte antigen (HLA) genes. These genes are central mediators of an immune response to not only a virus but also in pregnancy.  Some discussion about how these immune response genes may be influencing disease would be of great interest or at least mentioning the potential role of these genes in pregnancy and viral response would be an important element for discussion.

This additional section is now added to the manuscript:

3.2. Human Leukocyte Antigen (HLA)

An important topic to mention is the role of HLA genes in pregnancy when hosting the semi-allogenic fetus, and its effects on the maternal immune response. Immunologically tolerating the fetus is an essential phenomenon in pregnancy, of which HLA-C and HLA-G are some of the important genes involved in establishing this maternal-fetal immune intolerance [82, 83]. Firstly, paternal HLA-C has been shown to regulate CD4+ and CD8+ T lymphocyte activation through the expression of IL-10 and TGF-b [82]. Secondly, HLA-G is another crucial gene detected only in fetal cells at the maternal-fetal junction, unlike other class I genes which are expressed on nearly all nucleated cells [83]. The HLA-G isoforms contain a transmembrane a1-domain that is known to be capable of inhibiting NK cell activity which is thought to play a role in preventing trophoblast cell destruction by NK cells in the uterus where they are abundant [83]. HLA-G has also been shown to induce T lymphocyte death [84]. When considered together, the immunomodulatory effects of HLA-C and HLA-G in pregnancy may have further implications in the face of SARS-CoV-2 infection and possible increased susceptibility. However, since studies so far have mainly focused on the maternal-fetal interface, it is still unclear what the effects of HLA genes in pregnancy are on the systemic immune response. This may be a future area of exploration in investigating pregnancy state susceptibility to SARS-CoV-2 infection.
